# Development of a digital tool to assist in monitoring compliance for a public health initiative: "A Better Choice Food and Drink Supply Strategy for Queensland Healthcare Facilities"

Hai Pham [1], Sherridan Cluff[2], Mathew Dick[2], Erica Clifford[2], Nicole McDonald[1], Jason D. Pole [1,3]*

1 Queensland Digital Health Centre, The University of Queensland, Brisbane, Australia, 2 Health and Wellbeing Queensland, Brisbane, Australia, 3 Dalla Lana School of Public Health, the University of Toronto, Canada

* j.pole@utoronto.ca

## Abstract

### Background

Health and Wellbeing Queensland (Australia) is leading Queensland Government efforts to enhance food environments, ensuring that Queenslanders have access to healthy food and drinks options in places outside their homes. In healthcare settings, the *A Better Choice (ABC): Food and Drink Supply Strategy for Queensland Healthcare Facilities* is a policy that sets targets for the availability and promotion of food and drinks in retail outlets and vending machines with the aim of providing and promoting healthier options for staff and visitors. To strengthen policy action, a mandatory *Health Service Directive: Healthier food and drinks at healthcare facilities* (Directive) requires Hospital and Health Services (HHS) to meet ABC targets. Compliance with the Directive is assessed through annual audits.

### Objective

The objective of this study was to establish an enduring data and analytics solution to assist in the implementation of ABC and monitoring of compliance with the Directive.

### Methods

The Queensland Digital Health Centre, in collaboration with Health and Wellbeing Queensland developed a digital dashboard to report and manage audit data on food and drinks supplied in retail outlets and vending machines across Queensland public HHSs. Annual survey data was completed by 16 HHSs and Mater Health. These data were checked, aggregated and loaded into the digital analytics dashboard.

**Data availability statement:** All relevant data are within the paper and its Supporting Information files.

**Funding:** The project was funded by the Queensland Government through Health and Wellbeing Queensland, as part of The University of Queensland Health Research Accelerator initiative program for the Queensland Digital Health Centre (grant no. 2022000922).

**Competing interests:** The authors have declared that no competing interests exist.

## Results

The development process resulted in a replicable digital dashboard for reporting and decision-making. The ABC dashboard provides previous and current compliance data for all HHSs, featuring visualisations that illustrate changes in compliance over time to help identify emerging trends. Users can interact with the dashboard to filter data by HHS, year and by outlet and food or drink type. This digital innovation has facilitated faster delivery of food and drink supply trend analysis and compliance reporting for HHSs and statewide policy makers.

## Conclusions

Digital dashboards for public health policy compliance enable greater interrogation of data and provide visualisation tools to track trends in compliance over time. This allows more responsive and effective action to increase the impact of public health policies.

## Introduction

Digital dashboards play a crucial role in public health surveillance by providing real-time data visualisation and analysis. They enable health officials to monitor disease outbreaks [1], track vaccination rates [2], and assess the effectiveness of public health interventions [3]. By consolidating data from various sources into an easily interpretable format, digital dashboards facilitate quick decision-making and timely responses to emerging health issues. Additionally, if public, they enhance transparency and communication, ensuring that accurate information is readily available. Overall, digital dashboards are indispensable tools for improving public health outcomes and managing health crises efficiently.

Current literature highlights the significant role of digital health dashboards in the context of public health risks and diseases. A systematic review examining the state of research on dashboards in the context of public health risks and diseases found 65 studies reported the development of digital dashboards to monitor infectious diseases (e.g., Dengue, Corona, Ebola; N = 21), crises (e.g., disaster, emergencies; N = 10), health related services (e.g., aging population, health promotion programs; N = 17), and other health hazards (e.g., substance abuse, pollution; N = 17) [4]. The systematic review noted that while many studies focused on the functional aspects of dashboards, there was a need for more user-centric evaluations [4]. A scoping review published in 2024 identified key design principles for developing effective public health surveillance dashboards, emphasizing the importance of user requirements, robust infrastructure, and intuitive interfaces [5]. Examples of these elements include user requirements such as fast data access, customizable views, and minimal training shape dashboard design through filterable visuals, simple navigation, and role-specific access. Robust infrastructure ensures secure handling of large datasets and timely updates via scalable systems and reliable data pipelines. Intuitive interfaces reduce cognitive load with clean layouts, consistent icons, and interactive

features like tooltips and drilldowns (an interactive technique allowing users to see more detailed information when selecting a category).

Monitoring health policy compliance is essential to ensure that regulations and guidelines are effectively implemented, ultimately safeguarding and protecting public health. By tracking adherence to policies such as vaccination mandates, hygiene protocols, and disease reporting requirements, authorities can identify gaps and areas needing improvement. Digital dashboards facilitate this process by providing a centralised platform for real-time data collection and analysis. They allow health officials to visualise compliance trends, generate reports, and quickly identify non-compliance issues. This timely information enables targeted interventions and resource allocation, ensuring that health policies are enforced effectively, and public health goals are met.

Still, there is sparse evidence on the use of digital dashboards to monitor preventive health policies, particularly in the domain of nutrition and institutional food environments. While platforms such as the Australian Food Environment Dashboard [6], Global Food Environment Dashboard [7], and Food System Dashboard [8] demonstrate the growing use of interactive tools in public health nutrition, they primarily focus on population-level indicators, disease burden, or broader environmental metrics. These examples reflect a broader trend away from static, report-heavy outputs toward dynamic, user-driven platforms that support tailored data exploration – especially valuable for time-poor policymakers. Building on this momentum, we aim to present a case study for developing a visual digital dashboard to assist in monitoring a preventive health policy related to nutrition within Queensland hospitals and health services.

The A Better Choice (ABC): Food and Drink Supply Strategy for Queensland Healthcare Facilities (ABC) was introduced as a policy by Queensland Health in 2007 [9]. The aim of ABC is to improve the health and wellbeing of Queenslanders by increasing the availability and promotion of healthier food and drink options to staff and visitors within public healthcare settings [6]. The Queensland Hospital and Health Boards Act 2011 (Section 47) authorises the Chief Executive of the Department of Health to issue health service directives (HSD) to hospital and health services (HHSs) focused on specific requirements or outcomes. One such HSD mandates the implementation of ABC – Healthier food and drinks at healthcare facilities (the Directive) and was introduced for healthier drinks in 2019 and expanded to include healthier foods in 2020 [10]. Health and Wellbeing Queensland (HWQld), as the state's prevention agency, supports HHSs to implement ABC [11] to achieve the mandatory requirements outlined in the Directive.

## Materials and methods

### Study setting

ABC uses a traffic light system, based on the Australian Dietary Guidelines, to classify food and drinks as green (best nutritional value), amber (some nutritional value), or red (limit or no nutritional value) [10]. The Directive sets specific targets for retail outlets and vending machines (see S1 Table): for example, for food in retail outlet at least 50% of items must be green, no more than 20% may be red, and the remainder may be amber. Artificially sweetened (AS) beverages, such as diet soft drinks, are treated as a separate category due to their low energy content but potential influence on taste preferences and consumption behaviours. The specific target for AS drinks is not more than 20% of all drink options.

Queensland has 16 HHSs (geographically managed health system that are groups of hospitals under unified management) and Mater Health (a publicly funded hospital operating under its own act), which collectively provide public hospital and health services to a population of 5,215,000 people. In compliance with the Directive, Health providers (HHSs) are responsible for auditing the food and drinks supplied in retail outlets and vending machines within their facilities on an annual basis, which include a mix of charity, commercial and hospital operated outlets. This data is then utilised to assess adherence to the Directive.

## The "A Better Choice Dashboard" framework

Historically, reporting for the Healthier Food and Drinks at Healthcare Facilities Directive has relied on manual processes, with audit data collected by HHSs. Following data collection, HWQld is responsible for analysing the results, preparing PDF-based summary reports, and disseminating findings to individual HHSs. This process, while effective, has been time-intensive and dependent on static outputs that limit accessibility and real-time engagement. HWQld sought a collaboration with the Queensland Digital Health Centre (The University of Queensland) to explore areas for improvement and expansion in data input, data cleaning and report handling processes.

The first objective of this collaboration was to construct an interactive dashboard accessible to HWQld and potentially accessible by HHSs. This dashboard would permit deeper exploration of the data, paving the way to replace static PDF-based reports with access to digital data capable of greater levels of insights on performance and gaps. The second objective was to review the data reporting method. Originally, HHSs were required to provide data from half of their facilities and half of the outlets within those facilities, up to a maximum of eight outlets. Issues with this reporting process included yearly differences in the audited outlets (self-selected by HHSs) hindering trend monitoring and, in some cases, also limiting the visibility of outlet performance to those 'better performing' outlets selected by the HHS. The aim was to develop a data collection approach that captured all eligible outlets across a two-year period, whilst minimising reporting burden for large HHSs, streamlining the reporting process, and enabling timely monitoring of compliance.

The study was approved by the University of Queensland Human Research Ethics Committee (Ref: 2023/HE002184).

## Design and implementation

The dashboard was developed using the Microsoft Power Business Intelligence tool – a data visualization platform used for business intelligence purposes. It is a collection of software services, applications, and connectors that work together to turn data into coherent, visually immersive and interactive insights. This tool was chosen due to its widespread use across HHSs and HWQld, facilitating easier implementation and better access to the interactive dashboard.

## Data collection and sources

Prior to 2023, each HHS received a survey towards the end of the calendar year to report by the following January, the proportions of food and drinks from the green, red and amber categories, and artificially sweetened drinks, in their retail outlets and vending machines. This process has been in place for several years, and HHSs are familiar with the classification criteria outlined in the Healthier Food and Drinks Directive. In 2023, the survey was modified to report the number of items instead of proportions. This was to reduce the burden on the persons who complete the survey as requiring only simple counts and no calculations. Also introduced in 2023, the outlets needing to report were specified by HWQld to ensure that all retail outlets and vending machine brands within each HHS reported on their compliance within any two-year cycle. This reduced the risk of auditing being limited to the same selected outlets each year, with associated limited visibility of overall outlet performance. The 2023 survey data was captured using Qualtrics, an online survey tool used for teaching, learning, and research. After receiving the survey results back from HHSs, a team at HWQld checked the face validity of the data for each HHS. The data were then extracted into excel files and forwarded to the data analysis team at Queensland Digital Health Centre for processing.

## Data cleaning and compilation

Audit data from each HHS was imported into RStudio (version 4.4.0) for cleaning and manipulation. This process involved checking for data entry errors such as missing values and outliers or inconsistencies. The proportion for each category and outlet compliance was calculated based on the number of items reported for each category. The 2023 dataset was

then merged with historical audit data and the resulting cleaned and consolidated dataset was integrated into the digital dashboard.

## User interface and accessibility

The dashboard was designed to be user-friendly and in a form that could potentially be deployed to HHSs to stay informed about their compliance data. Its comprehensive nature and layout prioritise intuitive navigation, enabling users, regardless of technical expertise, to engage with the dashboard effectively. Key interactive features, including dropdown filters, hover-over tooltips, and responsive charts, facilitate dynamic data exploration without the need for prior training.

## Dashboard features and functionality

The ABC dashboard displays both past and current data on compliance with the Directive to promote healthier food and drinks for Queensland's 16 HHSs and Mater Health. It features visualisations that illustrate changes in compliance over time, aiding in trend tracking. Users can interact with the dashboard to filter data by HHS, year, outlet type, and food or drink category. We added a feature in the dashboard that allows compliance to be weighted by service volume for each facility. Service volume, which included the total number of patient admissions, emergency admissions, and outpatient services, was obtained from quarterly hospital activity reports (Jul-Sept 2023) publicly available on the Queensland Health website [12]. Calculating HHS compliance as an average across all outlets is sufficient for yearly reporting against the Directive; however, it cannot describe the impact of implementing this public health initiative. Given the differences in service volumes between hospitals, and hence the proportion of the population that is exposed to the food and drink options at a specific site, weighted data provides added insight into the population reach of the policy.

## Results

### Overview of the developed dashboard

A digital dashboard was developed that provides stakeholders the ability to visualise the compliance with the Directive across Queensland public HHSs. The main components of the dashboard are (1) Snapshot of compliance by audited year (Fig 1), (2) Compliance with targets for food and drinks from the green and red categories, and artificially sweetened drinks (Fig 2), (3) Water availability (Fig 3), (4) HHS performance data (Fig 4), and (5) Compliance trends across the years (Fig 5). All figures included for illustrative purposes were produced using synthetic data.

Fig 1 provides a snapshot of compliance for the selected year, including the number of facilities/outlets audited, compliance by categories such as food type, outlet type, HHS size, retail outlet owners, and vending machine brands. This dashboard page allows for a quick review of the overall compliance for each year. On page 2 of the dashboard (Fig 2), we present further details on compliance with the targets for food and drinks from the green and red categories, and artificially sweetened drinks, enabling users to identify which food/drink categories most frequently failed to meet the Directive requirements.

One of the assessment criteria for the Directive is the availability of free drinking water across HHSs, such as water fountains, in terms of being widely available, available in high traffic areas, or available but not easily accessible. The results of this assessment are presented in page 3 of the dashboard (Fig 3). The analysis was weighted by service volume, allowing users to observe the true proportion of Queenslanders with access to free drinking water across health facilities.

Fig 4 presents detailed compliance data for each HHS. This page replicates the PDF report previously sent to each HHS, allowing Executives to observe compliance data in full detail for each HSS outlet, and areas for improvement. We also introduced a near-compliance measurement to help identify those outlets and HHSs that are close to compliance, so

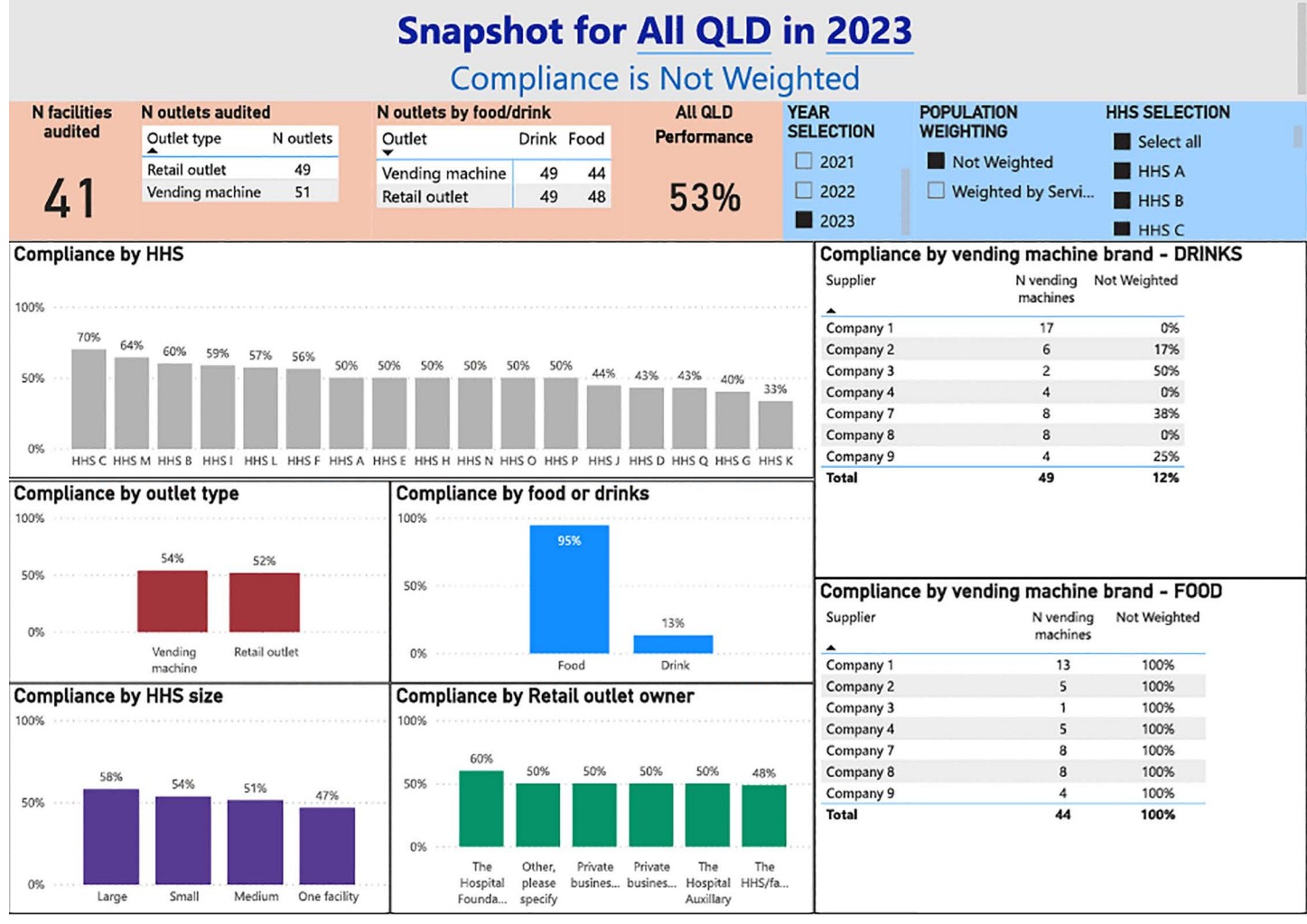

Note: Synthetic data was used for illustrative purposes.
The overall performance metric (e.g., 53%) for QLD in 2023 displayed in the dashboard represents the proportion of audited outlets that met the policy target. This metric provides a high-level summary of compliance across all participating HHSs. Detailed results are broken down by outlet type, food or drink, HHS size, retail outlet owner and vending machine supplier.

**Fig 1. Dashboard page 1. Snapshot by year.**

efforts can be focused on achieving full compliance. This page offers an option for electronic feedback to HHSs, replacing traditional static PDF reporting.

Fig 5 shows the compliance trend overtime, allowing users to track the changes in overall compliance, as well as averages of food and drinks from the green, red and amber categories and artificially sweetened drinks. In addition to these main pages, we created a flag for large changes (S1 Fig), which helps identify outlets or facilities that had significant changes from the previous year. Outlier flags were also created for outlets reporting very small or large numbers of items (S2 Fig); these thresholds were determined pragmatically, based on what is most useful for HWQld. These flagging functions support HWQld team – currently the primary user of the dashboard – in assessing data validity and initiate follow-up actions where potential data quality issues are identified.

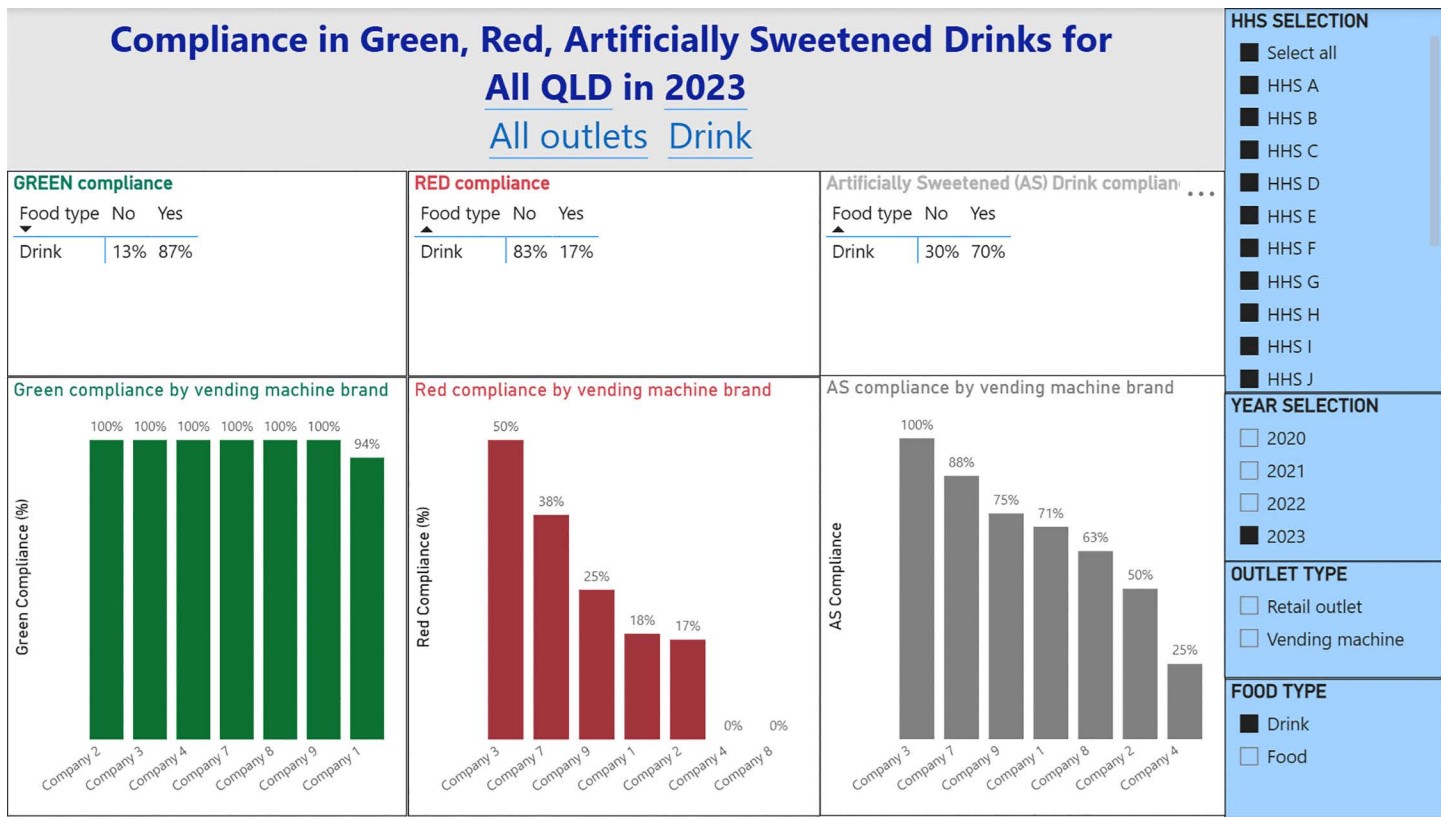

Note: Synthetic data was used for illustrative purposes.
The visualisation displayed in the page represent the proportions of audited outlet that met the policy target for each traffic light category (i.e., green, red, and artificially sweetened drink). Amber is not included in this page because there is no requirement for amber category in the policy to meet specific target.

**Fig 2. Dashboard page 2: Compliance in Green, Red, and artificially sweetened drinks.**

The dashboard development process has facilitated faster delivery of the food and drinks compliance report to HHSs and Queensland Health. The requirement for HWQ to manually clean and analyse data has been replaced by an automatic process, this assisted in the ability to send evidence-based outcomes reports to HHSs and Queensland Health in a timely manner.

## Discussion

This is the first study to discuss the development of a digital dashboard to visualise compliance with a public health directive across hospitals and health services in Australia. The ABC dashboard provides a clear and concise way to visualise compliance with the Directive, making it easier for policy makers to understand the trends and patterns of healthy food and drink supply in these facilities. It also provides real-time visualisations and useful information for data auditing and tracking.

We have successfully developed an interactive dashboard that is accessible to HWQld and shareable with HHSs to visualise compliance with the Directive. To date, there are no directly comparable systems that provide an accessible, policy-aligned, and visually structured overview of food environments within Australian hospitals and health services. While nutrition audit reports from jurisdictions such as Western Australia and New South Wales have offered valuable insights into compliance with healthy food policies, these reports are typically static and lack the interactive, user-oriented

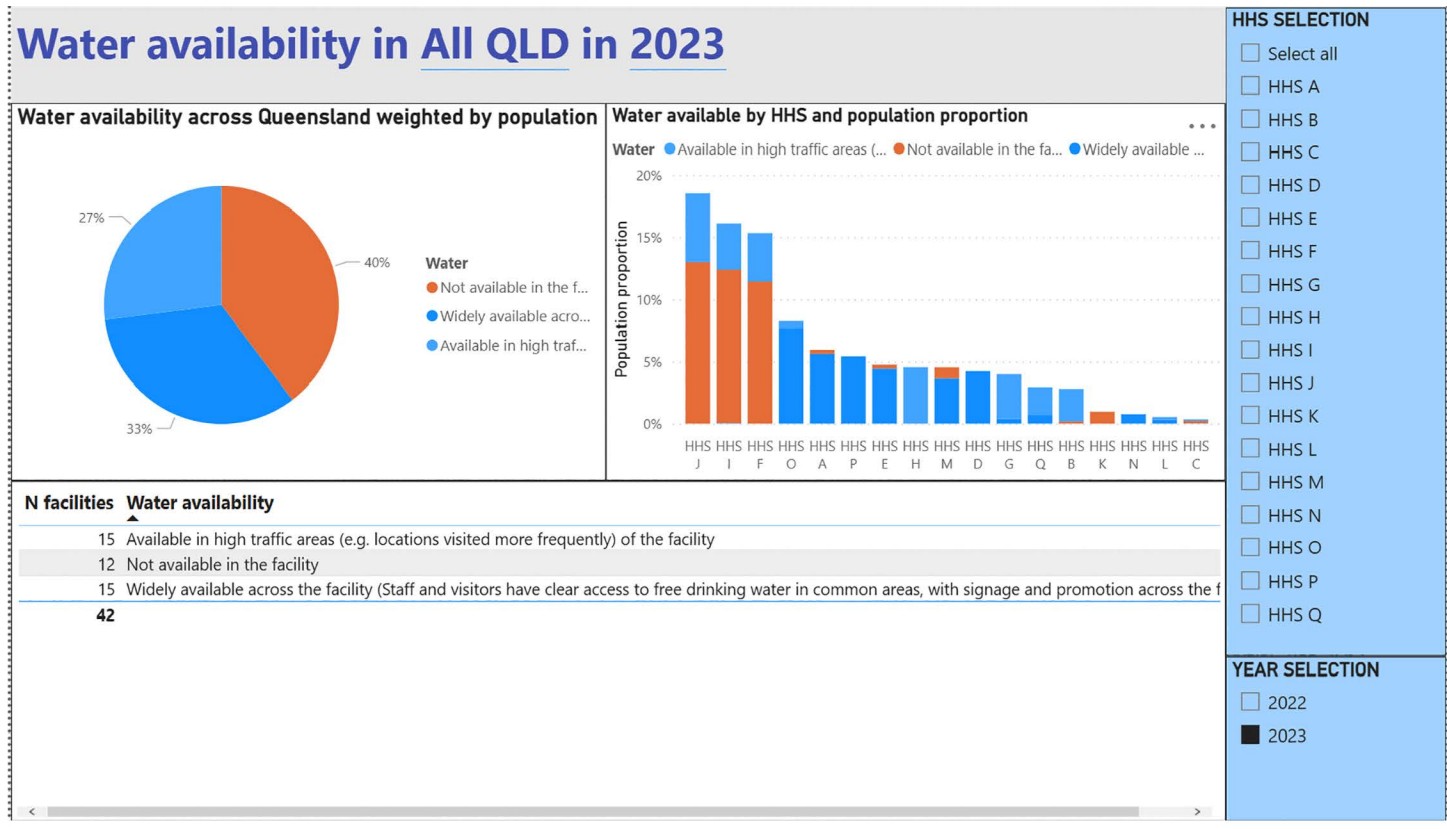

Note: Synthetic data was used for illustrative purposes.
The dashboard presents data on availability of free drinking water across participating HHSs, with results adjusted according to the service size of each HHS. While 30 out of 42 facilities (71.4%) reported that free drinking water was available or widely available, these facilities collectively serve only approximately 60% of the population represented in the dataset.

**Fig 3. Dashboard page 3: Water availability.**

design featured in this dashboard [6]. Compared to the traditional paper-based report, this new approach achieved faster turnaround times in providing feedback to HHSs due to streamlined survey structures and improved processes for data exporting, cleaning and analysis.

Implementing ABC presents several challenges including retailer resistance, limited capacity in HHSs to support implementation and maintaining leadership engagement and allocation of resources [13]. Given these challenges, continuous monitoring, reporting and implementation support are necessary to increase compliance with the Directive. The ABC dashboard provides the foundation for more efficient and detailed data interrogation and compliance tracking. This in turn, enables HWQld to communicate targeted messages to HHS Executives and focus implementation and support services to areas where greater effort is needed to improve compliance with the Directive.

Although this digital tool has improved many aspects of monitoring compliance, there are still a few areas with opportunities for further improvement. Firstly, maintaining data accuracy and quality is important to ensure accurate data is being used to inform policies. More resources and tools need to be allocated and investigated to ensure transparency and accuracy in data entry and data extraction. This includes provision of training to HHSs on when and how to enter data into the online survey and a timelier system for data checking and verification of the data supplied.

The dashboard is currently accessible to policy makers but not available to HHSs. The current design of the dashboard allows for a number of options for sharing; however, maintaining a robust digital dashboard requires significant technical

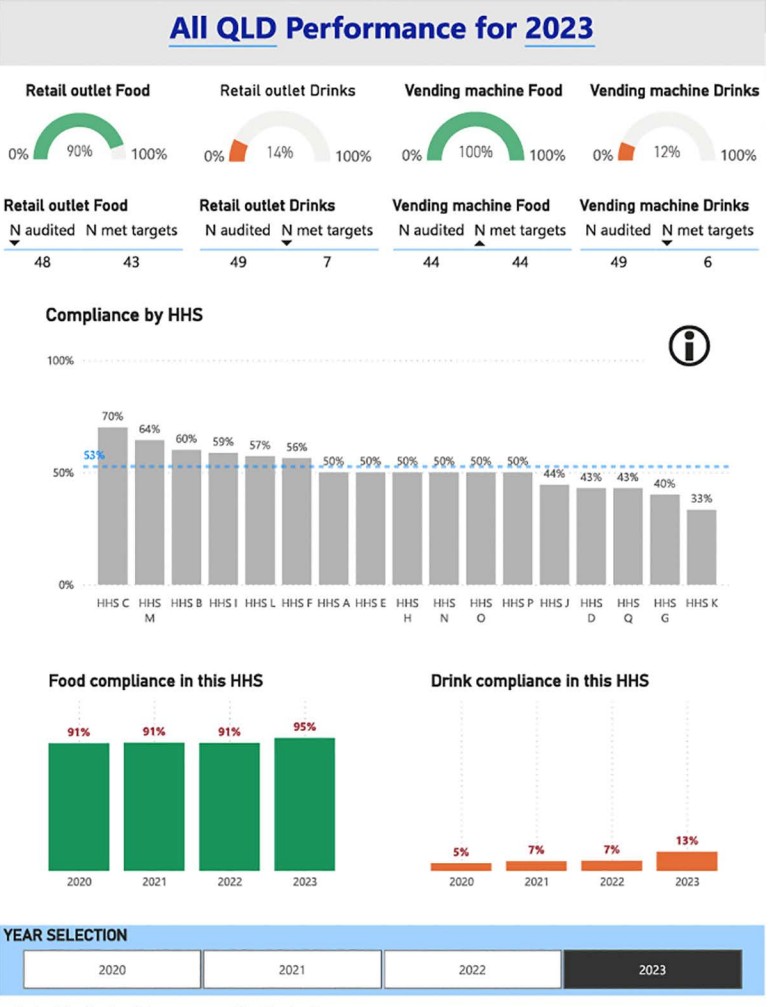

**Fig 4. Dashboard page 4: HHS compliance.**

expertise and infrastructure [5]. Ensuring the privacy and security of the data is critical. Effective use of dashboards requires training for users to interpret and use the data correctly [5]. All these aspects need to be considered before sharing the dashboards to broader users. Upcoming efforts will also focus on exploring effective methods for communicating results to individual retailers and determining which entities fall within the scope of the Directive. Future enhancements to the dashboard may include integrated statistical tools to analyse trend and identify differences in compliance rates.

While this dashboard has been developed specifically for Queensland, its design principles and monitoring framework have broader applicability. It is important to note that food and nutrition policies vary significantly across Australian states and territories, with each jurisdiction adopting distinct policy targets, implementation strategies, and compliance mechanisms. By aligning the dashboard with the Queensland Directive, we ensured contextual relevance; however, the underlying structure, particularly its modular design, service-volume weighting, and visual reporting tools, could be adapted to suit other jurisdictions across Australia and internationally. Acknowledging these policy differences not only clarifies the scope of the current tool but also highlights its potential as a scalable model for national or cross-jurisdictional monitoring of institutional food environments.

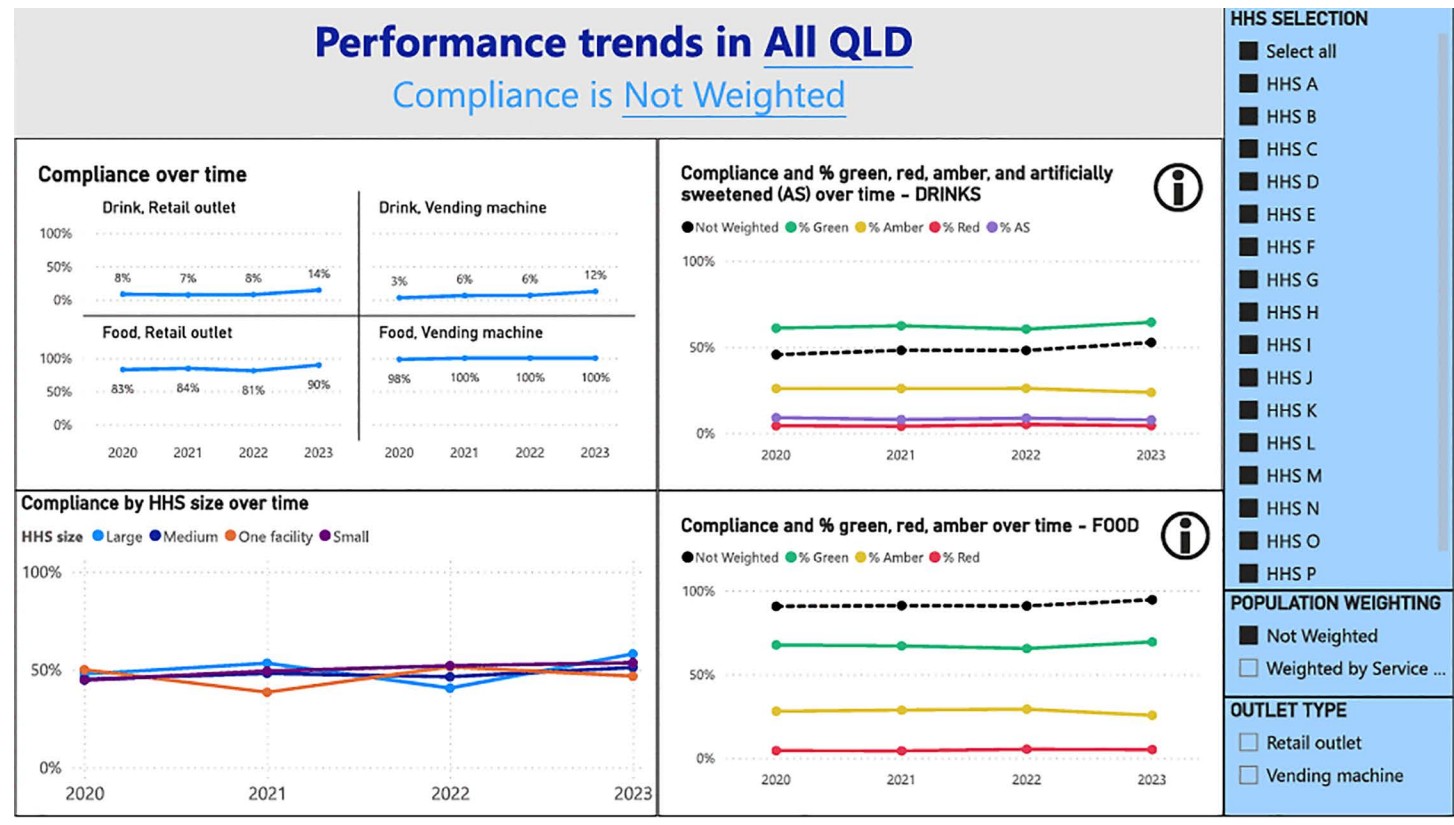

Note: Synthetic data was used for illustrative purposes.
This dashboard presents longitudinal trends in compliance and proportional distribution of items across the traffic light categories including green, amber, red, and AS drinks. As illustrated, there was a notable improvement in drink compliance within vending machines across Queensland, with a 9% increase observed between 2020 and 2023.

**Fig 5. Dashboard page 5: Compliance trends over the year.**

In conclusion, the development of a digital dashboard to visualise compliance with a food and drink supply strategy in retail outlets and vending machines across Queensland public HHSs has been demonstrated to be effective. The dashboard presented real-time tracking of compliance to a public health policy and provided visualisations for trends and patterns over time, which provides policy makers with more tools for informed decision-making. This data analysis supports targeted action to help HHSs continue improving the quality of food and drinks for staff and visitors, contributing to a healthier food environment that supports the health and wellbeing of Queenslanders.

## Supporting information

**S1 Table. Specific targets for retail outlets and vending machines food and drinks.**
(PDF)

**S1 Fig. Flags for large deviation.**
(TIF)

**S2 Fig. Additional flags for data validity check.**
(TIF)

## Acknowledgments

We acknowledge Ms Brooke Maund at Queensland Statewide Food Services for her support in updating data collection survey and communication of changes to HHSs regarding the new method of data collection.

## Author contributions

**Conceptualization:** Jason D. Pole.

**Data curation:** Hai Pham, Sherridan Cluff, Mathew Dick, Erica Clifford.

**Formal analysis:** Hai Pham, Jason D. Pole.

**Investigation:** Hai Pham, Sherridan Cluff, Mathew Dick, Erica Clifford, Jason D. Pole.

**Methodology:** Hai Pham, Jason D. Pole.

**Project administration:** Sherridan Cluff, Mathew Dick, Erica Clifford, Nicole McDonald.

**Supervision:** Jason D. Pole.

**Validation:** Hai Pham, Erica Clifford, Jason D. Pole.

**Visualization:** Hai Pham, Sherridan Cluff, Mathew Dick, Erica Clifford, Jason D. Pole.

**Writing – original draft:** Hai Pham.

**Writing – review & editing:** Sherridan Cluff, Mathew Dick, Erica Clifford, Nicole McDonald, Jason D. Pole.

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
