## [Decision Letter · Decision Letter 0]

1 Sep 2025

Dear Dr. Pole,

Thank you for submitting your manuscript to PLOS ONE. After careful consideration, we feel that it has merit but does not fully meet PLOS ONE’s publication criteria as it currently stands. Therefore, we invite you to submit a revised version of the manuscript that addresses the points raised during the review process.

We look forward to receiving your revised manuscript.

Kind regards,

Ali B. Mahmoud, Ph.D.

Academic Editor

PLOS ONE

Journal Requirements:

“The project was funded by the Queensland Government through Health and Wellbeing Queensland, as part of The University of Queensland Health Research Accelerator initiative program for the Queensland Digital Health Centre.”

“The project was funded by the Queensland Government through Health and Wellbeing Queensland, as part of The University of Queensland Health Research Accelerator initiative program for the Queensland Digital Health Centre.”

“The project was funded by the Queensland Government through Health and Wellbeing Queensland, as part of The University of Queensland Health Research Accelerator initiative program for the Queensland Digital Health Centre. We also acknowledge Dr Brooke Maund at Queensland Statewide Food Services for supporting in updating data collection survey and communication of changes to HHSs regarding the new method of data collection. “

“The project was funded by the Queensland Government through Health and Wellbeing Queensland, as part of The University of Queensland Health Research Accelerator initiative program for the Queensland Digital Health Centre.”

6. We note that you have indicated that there are restrictions to data sharing for this study. For studies involving human research participant data or other sensitive data, we encourage authors to share de-identified or anonymized data. However, when data cannot be publicly shared for ethical reasons, we allow authors to make their data sets available upon request. For information on unacceptable data access restrictions, please see http://journals.plos.org/plosone/s/data-availability#loc-unacceptable-data-access-restrictions.

Reviewers' comments:

Reviewer's Responses to Questions

**Comments to the Author**

1. Is the manuscript technically sound, and do the data support the conclusions?

Reviewer #1: Partly

Reviewer #2: Yes

2. Has the statistical analysis been performed appropriately and rigorously?

Reviewer #1: N/A

Reviewer #2: N/A

3. Have the authors made all data underlying the findings in their manuscript fully available?

Reviewer #1: No

Reviewer #2: Yes

4. Is the manuscript presented in an intelligible fashion and written in standard English?

Reviewer #1: Yes

Reviewer #2: Yes

Reviewer #1: The work require a mix of professionals and a high level of IT involvement. A considerable work has been done

User interface and accessibility: (I) how user-friendly it is? Any advantage in relation to other systems used earlier? Interactivity????

(ii) The idea compliance is weighted by service volume for each facility is not clear, because I could not see relationship between food and drink provided and compliance. The principle, I thought, is compliance regardless of the size of the institution served.

In the Discussion section, authors could not compare the system with any other system which may have similar function. If this is completely new, it is good to say it loudly. Even no comparison was made with the previous system to show the relative advantage of the current one.

It would have been good to also use the data for some statistical analysis to show it on the dashboard

Reviewer #2: Development of a digital tool to assist in monitoring compliance for a public health initiative: "A Better Choice Food and Drink Supply Strategy for Queensland Healthcare Facilities"

Thank you for the opportunity to review this manuscript. It is a timely and engaging contribution to the field, particularly in its innovative approach to visualising policy-related data on food and drink availability in Queensland. The development of a dashboard tool to support both the interpretation of results and the monitoring of compliance with the ABC Policy is a valuable initiative that uses existing platforms and software. The manuscript is generally well-structured and clearly written. Below is a series of comments that aim to support further refinement of the manuscript.

Scope and Policy Context

The manuscript appropriately focuses on Queensland, which is appropriate given the jurisdictional differences in policy targets across Australian states. A brief note acknowledging that different states have varying policies and policy targets could enhance the clarity for readers outside Australia, and indicate whether the tool could be adapted for use in other jurisdictions. This would help readers understand the broader relevance of the approach.

Introduction

The introduction outlines several monitoring systems, but it would be strengthened by the inclusion of more examples directly related to nutrition and food environments. Relevant dashboards include the World Obesity Observatory (https://data.worldobesity.org/), the GBD Compare tool (https://vizhub.healthdata.org/gbd-compare/), and the Australian Food Environment Dashboard (https://foodenvironmentdashboard.com.au/) - but others can also be included. Briefly summarising the focus of such platforms would situate the study within ongoing public health nutrition initiatives and underline the rationale for the proposed dashboard (even if only available internally). These examples also reflect a broader trend away from static, report-heavy outputs towards interactive, user-driven platforms that allow tailored exploration of data (especially for time-poor policy makers).

The first paragraph on page 4 briefly outlines findings from the scoping review. However, this section would benefit from examples clarifying what "user requirements, robust infrastructure and intuitive interfaces" entail in practice. For instance, how such requirements shape dashboard design and usability.

NB Page 5 paragraph 1: There appears to be "(ref)" included after (8), possibly a typo.

Clarifications

The manuscript refers to traffic light food and drink categories. On page 5, paragraph 2, 'red' items should it say (limited or no nutritional value)? It would be helpful to include a brief explanation of what specific targets or thresholds apply under the policy to the green, amber and red categories, and why "artificially sweetened beverages" are treated differently as a separate category (and what is the target for these)? This would improve accessibility for readers unfamiliar with the policy framework.

The term "Mater Health" is used without explanation. A brief clarification in brackets or a footnote would be helpful for readers unfamiliar with Queensland's health system structure.

Please include the ref number for the ethics approval.

Purpose of the Dashboard and Aim of the Study

The manuscript states that the purpose of the dashboard is to monitor compliance/trends. However, it is more of a visualising tool for results and trends since the monitoring includes more broadly all actions taken to obtain the results (which does not diminish its value). The authors should use the terminology more consistently across the different parts of the manuscript. Additionally, the objective includes "to explore its usability for decision-makers"; it appears the study did not fully realise this aim - it would be expected that some interaction with policy makers took place to explore that (if this was the case, there might not be enough details included to explain this aspect).

It would be helpful to outline how reporting for this policy has been undertaken to date. Page 6 refers to PDF-based reports, but a clearer description of the existing process would add context; for example, who is responsible for analysing the data after collection, preparing the reports, and disseminating them. If available, information on how the proposed dashboard approach compares in terms of streamlining or feasibility would also strengthen the case for its added value.

Data Preparation and Analysis

The manuscript would benefit from a more detailed description of the data flow between Qualtrics and the dashboard. Key points needing clarification include what data are collected in hospitals (e.g. product photos, nutrition information), how items are classified as green/amber/red (and at what point), who undertakes data cleaning and quality assurance, how discrepancies across services are resolved, and how was data that previously was reported as percentages integrated with the updated reporting on numbers. Greater detail would help readers understand the workflow and effort required to generate dashboard outputs for those wanting to develop their own dashboard.

Dashboard

The functionality of the dashboard is well outlined, and the weighted outputs by population are a particularly strong feature, demonstrating the potential of a digital approach that integrates data from multiple sources. This aspect is well done. A few points could be clarified further:

-Figures: In Figure 2, there are no amber results, whereas amber appears in Figure 4; an explanation would be helpful.

-Flags: More detail is needed on how the dashboard flags specific items, why certain flags were chosen, and how these flags support validity checks, given that most users would not independently verify data quality.

-Communication: It would also be useful to describe how results are (or will be) communicated back to individual retailers (if at all), as this will be important for practical application.

Figures

The mock-up figures illustrating dashboard results are generally self-explanatory, but some features would benefit from additional clarification, for example, through a legend. It is not clear what the 53% value shown for the overall performance metric represents (and for the other, more detailed results), compared to the green/amber/red categories. A greater explanation of how users or policymakers should interpret such metrics would improve clarity, and where the flagging function would be shown.

**Do you want your identity to be public for this peer review?** For information about this choice, including consent withdrawal, please see our Privacy Policy

Reviewer #1: **Yes:** Girma Taye

Reviewer #2: No

---

## [Author Response · Author response to Decision Letter 1]

3 Nov 2025

Reviewer 1:

The work require a mix of professionals and a high level of IT involvement. A considerable work has been done.

Thank you for acknowledging the interdisciplinary nature and technical complexity of this project.

1. User interface and accessibility: (I) how user-friendly it is? Any advantage in relation to other systems used earlier? Interactivity????

Thank you for raising these important points regarding the dashboard’s usability and interactivity. As stated in the manuscript, we used Power BI to develop the dashboard which is widely accessible via desktop, web, and mobile apps and widely used across HWQld and HHSs. We designed the user interface with a strong emphasis on simplicity, clarity, and interactivity. The layout ensures that users – regardless of technical background – can intuitively navigate the dashboard. Key features such as dropdown filters, hover-over tooltips, and responsive charts allow users to explore data dynamically without requiring prior training.

Comparing to other systems such as the Australian food environment dashboard (foodenvironmentdashboard.com.au) which reports food availability across hospital and heath services in Western Australia and New South Wales along with compliance with government policy and recommendations, our dashboard provides detailed audit information in a user-friendly and highly interactive way. Our dashboards inclusion of comprehensive data visualisation and interactivity are key design features that set it apart.

To highlight this, the advantages of user interface and interactivity have been added to the manuscript in the method and discussion sections (paragraph 3, page 8 and paragraph 4 page 11).

“The dashboard was designed to be user-friendly and in a form that could potentially be deployed to HHSs to stay informed about their compliance data. Its comprehensive nature and layout prioritise intuitive navigation, enabling users, regardless of technical expertise, to engage with the dashboard effectively. Key interactive features, including dropdown filters, hover-over tooltips, and responsive charts, facilitate dynamic data exploration without the need for prior training.”

“We have successfully developed an interactive dashboard that is accessible to HWQld and shareable with HHSs to monitor compliance with the Directive. To date, there are no directly comparable systems that provide an accessible, policy-aligned, and visually structured overview of food environments within Australian hospitals and health services. While nutrition audit reports from jurisdictions such as Western Australia and New South Wales have offered valuable insights into compliance with healthy food policies, these reports are typically static and lack the interactive, user-oriented design featured in this dashboard (6). Compared to the traditional paper-based report, this new approach achieved faster turnaround times in providing feedback to HHSs due to streamlined survey structures and improved processes for data exporting, cleaning and analysis.”

2. The idea compliance is weighted by service volume for each facility is not clear, because I could not see relationship between food and drink provided and compliance. The principle, I thought, is compliance regardless of the size of the institution served.

The decision to apply service volume weighting was based on the principle of equity and proportional impact. Facilities that serve a greater volume of the public and hence a larger number of meals or beverages have a greater influence on the overall food environment within the health system. Weighting compliance scores by service volume ensures that the evaluation reflects not just whether healthy options are available, but how widely they are accessed. This approach captures the real-world exposure of consumers to healthier (or less healthy) choices, giving greater analytical weight to facilities that shape dietary patterns at scale. In doing so, it supports a more equitable and representative assessment of system-wide food service performance.

For example, a small facility with high compliance may have limited reach, whereas a large hospital with low compliance could affect thousands of consumers daily. Weighting ensures that the dashboard captures this difference in scale and better represents the population-level implications of food environment policies.

That said, we acknowledge that this approach may not be immediately intuitive and recognize the value of presenting both unweighted compliance alongside weighted to allow users to interpret both perspectives. No changes were made to the manuscript.

3. In the Discussion section, authors could not compare the system with any other system which may have similar function. If this is completely new, it is good to say it loudly. Even no comparison was made with the previous system to show the relative advantage of the current one.

To our knowledge, there are no directly comparable systems that offer an accessible, policy-aligned, and visually structured overview of food environments within Australian hospitals and health services. While nutrition audit reports such as those conducted by the Western Australia Department of Health and New South Wales Health provide valuable data on compliance with healthy food policies, these are typically presented in static formats and lack the interactive, user-friendly interface offered by the Food Environment Dashboard (foodenvironmentdashboard.com.au). We have now emphasised this advantage in the manuscript as described in response 1 (paragraph 4, page 11).

4. It would have been good to also use the data for some statistical analysis to show it on the dashboard

Statistical analysis is only useful if there is a specific question of difference that is trying to be assessed. For this project, there were no questions of difference proposed that required a statistical analysis. That said, we thank you for this suggestion and agree that incorporating statistical analysis using real audit data could strengthen the demonstration of the dashboard’s analytical capabilities when appropriate. We have added in paragraph 3, page 12 a statement to integrate statistical tools to analyse trend and identify differences in compliance rates across HHSs as one of the upcoming focus areas.

Paragraph 4, page 12 now reads:

“The dashboard is currently accessible to policy makers but not available to HHSs. The current design of the dashboard allows for a number of options for sharing; however, maintaining a robust digital dashboard requires significant technical expertise and infrastructure [5]. Ensuring the privacy and security of the data is critical. Effective use of dashboards requires training for users to interpret and use the data correctly [5]. All these aspects need to be considered before sharing the dashboards to broader users. Upcoming efforts will also focus on exploring effective methods for communicating results to individual retailers and determining which entities fall within the scope of the Directive. Future enhancements to the dashboard may include integrated statistical tools to analyse trend and identify differences in compliance rates.”

Reviewer 2:

Development of a digital tool to assist in monitoring compliance for a public health initiative: "A Better Choice Food and Drink Supply Strategy for Queensland Healthcare Facilities"

Thank you for the opportunity to review this manuscript. It is a timely and engaging contribution to the field, particularly in its innovative approach to visualising policy-related data on food and drink availability in Queensland. The development of a dashboard tool to support both the interpretation of results and the monitoring of compliance with the ABC Policy is a valuable initiative that uses existing platforms and software. The manuscript is generally well-structured and clearly written. Below is a series of comments that aim to support further refinement of the manuscript.

1. Scope and Policy Context

The manuscript appropriately focuses on Queensland, which is appropriate given the jurisdictional differences in policy targets across Australian states. A brief note acknowledging that different states have varying policies and policy targets could enhance the clarity for readers outside Australia and indicate whether the tool could be adapted for use in other jurisdictions. This would help readers understand the broader relevance of the approach.

Thank you for this thoughtful suggestion. We agree that acknowledging the variation in food and nutrition policies across Australian states would enhance clarity, particularly for international readers. In response, we have revised the Discussion section to include a brief note highlighting these jurisdictional differences and the implications for dashboard adaptability. While the current dashboard is tailored to Queensland’s Directive, its modular structure, policy-aligned metrics, and scalable design allow for adaptation to other jurisdictions with similar institutional food environment policies. This addition helps contextualize the broader relevance of the tool and its potential application beyond Queensland.

Paragraph 2, page 13 now reads:

“While this dashboard has been developed specifically for Queensland, its design principles and monitoring framework have broader applicability. It is important to note that food and nutrition policies vary significantly across Australian states and territories, with each jurisdiction adopting distinct policy targets, implementation strategies, and compliance mechanisms. By aligning the dashboard with the Queensland Directive, we ensured contextual relevance; however, the underlying structure—particularly its modular design, service-volume weighting, and visual reporting tools—could be adapted to suit other jurisdictions across Australia and internationally. Acknowledging these policy differences not only clarifies the scope of the current tool but also highlights its potential as a scalable model for national or cross-jurisdictional monitoring of institutional food environments.”

2. Introduction

The introduction outlines several monitoring systems, but it would be strengthened by the inclusion of more examples directly related to nutrition and food environments. Relevant dashboards include the World Obesity Observatory (https://data.worldobesity.org/), the GBD Compare tool (https://vizhub.healthdata.org/gbd-compare/), and the Australian Food Environment Dashboard (https://foodenvironmentdashboard.com.au/) - but others can also be included. Briefly summarising the focus of such platforms would situate the study within ongoing public health nutrition initiatives and underline the rationale for the proposed dashboard (even if only available internally). These examples also reflect a broader trend away from static, report-heavy outputs towards interactive, user-driven platforms that allow tailored exploration of data (especially for time-poor policy makers).

Thank you for this valuable suggestion. We agree that situating our dashboard within the broader landscape of public health nutrition monitoring tools will enhance the relevance and clarity of the Introduction. In response, we have revised the section to include examples of established platforms such as the Australian Food Environment Dashboard, Global Food Environment Dashboard, and Food System Dashboard. These examples illustrate the growing shift toward interactive, user-driven systems that support tailored data exploration—particularly beneficial for policy makers and practitioners. This addition helps contextualize our dashboard within ongoing global and national efforts to improve food environments and public health outcomes.

Paragraph 3, page 4 now reads:

“Still, there is sparse evidence on the use of digital dashboards to monitor preventive health policies, particularly in the domain of nutrition and institutional food environments. While platforms such as the Australian Food Environment Dashboard (6), Global Food Environment Dashboard (7), Food System Dashboard (8) demonstrate the growing use of interactive tools in public health nutrition, they primarily focus on population-level indicators and broader environmental metrics. These examples reflect a broader trend away from static, report-heavy outputs toward dynamic, user-driven platforms that support tailored data exploration—especially valuable for time-poor policymakers. Building on this momentum, we aim to present a case study for developing a visual digital dashboard to monitor a preventive health policy related to nutrition within Queensland hospitals and health services”.

3. The first paragraph on page 4 briefly outlines findings from the scoping review. However, this section would benefit from examples clarifying what "user requirements, robust infrastructure and intuitive interfaces" entail in practice. For instance, how such requirements shape dashboard design and usability.

NB Page 5 paragraph 1: There appears to be "(ref)" included after (8), possibly a typo.

Thank you for citing the typo and also providing this helpful suggestion. We agree that providing concrete examples of how user requirements, infrastructure, and interface design influence dashboard development would enhance clarity and relevance. In response, we have revised the paragraph to include practical illustrations drawn from the scoping review and our own dashboard experience. These additions help demonstrate how such considerations translate into design decisions that improve usability, accessibility, and functionality for end users.

Paragraph 1 page 4 now reads:

“Current literature highlights the significant role of digital health dashboards in the context of public health risks and diseases. A systematic review examining the state of research on dashboards in the context of public health risks and diseases found 65 studies reported the development of digital dashboards to monitor infectious diseases (e.g., Dengue, Corona, Ebola; N=21), crises (e.g., disaster, emergencies; N=10), health related services (e.g., aging population, health promotion programs; N=17), and other health hazards (e.g., substance abuse, pollution; N=17) [4]. The systematic review noted that while many studies focused on the functional aspects of dashboards, there was a need for more user-centric evaluations [4]. A scoping review published in 2024 identified key design principles for developing effective public health surveillance dashboards, emphasizing the importance of user requirements, robust infrastructure, and intuitive interfaces [5]. Examples of these elements include user requirements such as fast data access, customizable views, and minimal training shape dashboard design through filterable visuals, simple navigation, and role-specific access. Robust infrastructure ensures secure handling of large datasets and timely updates via scalable systems and reliable data pipelines. Intuitive interfaces reduce cognitive load with clean layouts, consistent icons, and interactive features like tooltips and drilldowns.”

4. Clarifications

The manuscript refers to traffic light food and drink categories. On page 5, paragraph 2, 'red' items should it say (limited or no nutritional value)? It would be helpful to include a brief explanation of what specific targets or thresholds apply under the policy to the green, amber and red categories, and why "artificially sweetened beverages" are treated differently as a separate category (and what is the target for these)? This would improve accessibility for readers unfamiliar with the policy framework.

Thank you for this helpful suggestion. We agree that clarifying the traffic light classification system and associated targets will improve accessibility for readers unfamiliar with the policy. In response, we have revised the manuscript to define the green, amber, and red categories more explicitly, including the nutritional rationale behind each classification. We have also added a brief explanation of the specific thresholds applied to each category under the policy and clarified why artificially sweetened beverages are treated as a separate category, including the volume restriction that applies (Table S1). These additions aim to enhance clarity and contextual understanding for a broader audience

---

## [Decision Letter · Decision Letter 1]

19 Dec 2025

Dear Dr. Pole,

We look forward to receiving your revised manuscript.

Kind regards,

Ali B. Mahmoud, Ph.D.

Academic Editor

PLOS One

Journal Requirements:

Reviewers' comments:

Reviewer's Responses to Questions

**Comments to the Author**

Reviewer #1: All comments have been addressed

Reviewer #2: All comments have been addressed

2. Is the manuscript technically sound, and do the data support the conclusions?

Reviewer #1: Yes

Reviewer #2: Yes

3. Has the statistical analysis been performed appropriately and rigorously?

Reviewer #1: Yes

Reviewer #2: N/A

4. Have the authors made all data underlying the findings in their manuscript fully available?

Reviewer #1: No

Reviewer #2: Yes

5. Is the manuscript presented in an intelligible fashion and written in standard English?

Reviewer #1: Yes

Reviewer #2: Yes

Reviewer #1: Although I am satisfied by the responses to my question, I still feel that you need to correct the editorial and language issues in order to make the manuscript readable. marking of graph axis, their relative size, and the need for each of them need to be evaluated.

Reviewer #2: Thank you very much for addressing all of my previous comments and providing clarifications. I just have a few minor suggestions for further improvement:

Abstract - Results section:

The third sentence and the last sentence feel slightly repetitive, and could be reworded for clarity. It might also help to combine the entire visualisation description into one cohesive statement at the end of this section.

Abstract - Conclusion:

There is a conflict between singular and plural grammar. Suggest changing to a plural form as it is a conclusion so it refers to more generalisable findings.

Introduction - Second paragraph:

Briefly explain what 'drilldowns' are as features of the digital tools (e.g., in brackets) for reader clarity.

Materials and Methods - First paragraph:

In the last sentence, where it says 'AS drinks is no more than 20%', please clarify whether this refers to available drink options only or all options (including food and drink).

Consistency in terminology:

Throughout the text, there is now some inconsistency in describing the purpose of the digital dashboard (e.g., 'to monitor compliance' vs. 'to visualise compliance').

**Do you want your identity to be public for this peer review?** For information about this choice, including consent withdrawal, please see our Privacy Policy

Reviewer #1: **Yes:** Girma Taye Aweke

Reviewer #2: **Yes:** Magda Rosin

---

## [Author Response · Author response to Decision Letter 2]

6 Jan 2026

STATEMENT ADDRESSING REVIEWERS’ COMMENTS

Note: Responses to reviewers’ comments are in blue. Changes made to the manuscript are in underlined red. Page and paragraph numbers cited herein pertain to the marked-up version provided.

Reviewer #1:

Although I am satisfied by the responses to my question, I still feel that you need to correct the editorial and language issues in order to make the manuscript readable. marking of graph axis, their relative size, and the need for each of them need to be evaluated.

Thank you for your comments. We have revised the manuscript, corrected grammar issues, and reviewed the overall readability. We hope this has improved reliability but are happy to address any further specific feedback.

We have also reviewed the figures and appreciate the suggestions regarding axis labelling and relative sizing. However, the visualisations presented in this study were designed for use within an interactive dashboard intended for policy makers and stakeholders. As such, the standards applied to static figures in published materials may not be directly applicable. The dashboard visualisations allow users to interact with the data and access additional detail, which cannot be fully represented in static screenshots included in the manuscript.

Reviewer #2:

Thank you very much for addressing all of my previous comments and providing clarifications. I just have a few minor suggestions for further improvement:

1. Abstract - Results section:

The third sentence and the last sentence feel slightly repetitive, and could be reworded for clarity. It might also help to combine the entire visualisation description into one cohesive statement at the end of this section.

The results section of the abstract (on page 3) has been reworded for clarity and now reads:

“The development process resulted in a replicable digital dashboard for reporting and decision-making. The ABC dashboard provides previous and current compliance data for all HHSs, featuring visualisations that illustrate changes in compliance over time to help identify emerging trends. Users can interact with the dashboard to filter data by HHS, year and by outlet and food or drink type. This digital innovation has facilitated faster delivery of food and drink supply trend analysis and compliance reporting for HHSs and statewide policy makers.”

2. Abstract - Conclusion:

There is a conflict between singular and plural grammar. Suggest changing to a plural form as it is a conclusion so it refers to more generalisable findings.

Grammar has been checked and changed to plural form. The abstract conclusions section now reads (page 3):

“Digital dashboards for public health policy compliance enable greater interrogation of data and provide visualisation tools to track trends in compliance over time. This allows more responsive and effective action to increase the impact of public health policies.”

3. Introduction - Second paragraph:

Briefly explain what 'drilldowns' are as features of the digital tools (e.g., in brackets) for reader clarity.

“Drilldowns” function is now explained in more detailed in paragraph 2 of the Introduction:

“…Robust infrastructure ensures secure handling of large datasets and timely updates via scalable systems and reliable data pipelines. Intuitive interfaces reduce cognitive load with clean layouts, consistent icons, and interactive features like tooltips and drilldowns (an interactive technique allowing users to see more detailed information when selecting a category).”

4. Materials and Methods - First paragraph:

In the last sentence, where it says 'AS drinks is no more than 20%', please clarify whether this refers to available drink options only or all options (including food and drink).

Thank you for the suggestion, we have now clarified that the specific target for AS drinks is only for drink options:

“…The specific target for AS drinks is not more than 20% of all drink options.”

5. Consistency in terminology:

Throughout the text, there is now some inconsistency in describing the purpose of the digital dashboard (e.g., 'to monitor compliance' vs. 'to visualise compliance').

We acknowledge the inconsistency in how the purpose of the digital dashboard was described. We have revised the manuscript accordingly and now consistently state that the dashboard is designed to visualise, rather than monitor, compliance.

---

## [Decision Letter · Decision Letter 2]

12 Jan 2026

Development of a digital tool to assist in monitoring compliance for a public health initiative: “A Better Choice Food and Drink Supply Strategy for Queensland Healthcare Facilities”

PONE-D-25-19154R2

Dear Dr. Pole,

We’re pleased to inform you that your manuscript has been judged scientifically suitable for publication and will be formally accepted for publication once it meets all outstanding technical requirements.

Kind regards,

Ali B. Mahmoud, Ph.D.

Academic Editor

PLOS One

Additional Editor Comments (optional):

Reviewers' comments:

Reviewer's Responses to Questions

**Comments to the Author**

Reviewer #2: All comments have been addressed

2. Is the manuscript technically sound, and do the data support the conclusions?

Reviewer #2: Yes

3. Has the statistical analysis been performed appropriately and rigorously?

Reviewer #2: N/A

4. Have the authors made all data underlying the findings in their manuscript fully available?

Reviewer #2: Yes

5. Is the manuscript presented in an intelligible fashion and written in standard English?

Reviewer #2: Yes

Reviewer #2: (No Response)

**Do you want your identity to be public for this peer review?** For information about this choice, including consent withdrawal, please see our Privacy Policy

Reviewer #2: No

---

## [Editor Report · Acceptance letter]

PONE-D-25-19154R2

PLOS One

Dear Dr. Pole,

I'm pleased to inform you that your manuscript has been deemed suitable for publication in PLOS One. Congratulations! Your manuscript is now being handed over to our production team.

Kind regards,

on behalf of

Dr. Ali B. Mahmoud

Academic Editor

PLOS One